# Cognitive Diversity as the Quality of Leadership in Crisis: Team Performance in Health Service during the COVID-19 Pandemic

**DOI:** 10.3390/healthcare9030313

**Published:** 2021-03-11

**Authors:** Zuzana Joniaková, Nadežda Jankelová, Jana Blštáková, Ildikó Némethová

**Affiliations:** Department of Management, Faculty of Business Management, University of Economics in Bratislava, Dolnozemská Cesta 1, 852 35 Bratislava, Slovakia; nadezda.jankelova@euba.sk (N.J.); jana.blstakova@euba.sk (J.B.); ildiko.nemethova@euba.sk (I.N.)

**Keywords:** leadership, efficiency, cognitive diversity, decision-making, COVID-19, pandemics

## Abstract

The level of leadership skills of healthcare team leaders has long been the subject of interest and many discussions. Several studies have pointed to their inadequacy, which is becoming a serious problem during the global crisis due to the Covid-19 pandemic. There is a direct link between the leadership in the healthcare system and its performance, conditioned by the level of decisions of leaders of medical teams. It is they who determine the performance of healthcare delivery. The study published in this article contains the results from the examination of the dependence between crisis leadership and team performance in healthcare providers. The subject of the research is the impact of cognitive diversity and the quality of crisis-leadership decision-making on the performance of medical teams in the acute crisis phase. The study was conducted on a research sample of 216 healthcare providers after the outbreak of the COVID-19 pandemic in Slovakia (April 2020). The respondents to the research sample involved team leaders in healthcare providers, who have been involved in managing the crisis. The study has justified the positive association between crisis leadership and team performance, which is mediated by cognitive diversity, supporting the quality of decision-making in crisis leadership. The results of the research have proven that the performance of the medical team in the acute crisis phase can be positively influenced through qualified decision-making in crisis leadership amplified by the usage of cognitive diversity.

## 1. Introduction

The current crisis has emerged unexpectedly and caught the management of healthcare providers largely unprepared. The pandemic has generated unprecedented challenges in modern history, for which there is a lack of vigorous studies and procedures that could be directly applied. Leaders of healthcare providers face the urgent challenge of managing the performance of teams of healthcare professionals during a crisis to provide healthcare at its full potential adequate to the time of a pandemic. The human factor has long been talked about as the main resource that determines the efficiency of organizations [1]. The influence of leadership quality on employees’ performance has been proven in many studies. Effective leadership has one of the most important impacts on the effective and quality results. Therefore, the quality of leadership is closely related to the quality of healthcare service and the leadership is considered to be the base stone of well-coordinated and integrated healthcare providers [2]. There have been studies publishing results on significantly positive effects between leadership style and high satisfaction of patients and the decrease of negative impacts [3,4,5]. The relevant research gap is in the dimensions of leadership competence in times of crises. Competent leadership has the key role in handling the challenges of pandemics in healthcare providers. The importance of leadership competence of employees with managerial responsibility is currently the subject of discourse, but in practice it is often underestimated. Many authors point out the need to develop not only clinical but also leadership skills during postgraduate medical education [6,7,8,9]. The investment into leadership competence development, which needs to be approached as a holistic concept [6] should be involved in the strategy of healthcare providers with the intention to improve the quality of their services [8].

The crisis is creating non-standard and complex conditions in which adequate leadership skills are doubly required due to the need for quick and considered decisions not only with a short-term but also with a long-term impact [10]. In the acute phase of change, the performance of employees is affected by increased uncertainty. Uncertainty, as the most unsustainable state of the human mind, is a source of stress, which creates barriers to employee performance during a crisis. The aim of this study is to examine the content of leadership competence in healthcare providers in crises caused by COVID-19 pandemic and its relation to medical teams’ performance; as well as the mechanism, which mediates their common effect. Leadership in this study is understood as the competence of leaders, who have led the healthcare facility in crises. The topic of our study is important for several reasons. The leadership competence of leaders in healthcare providers, under conditions of pandemic has not been sufficiently examined.

Many studies have examined the content of leadership quality and pointed to the importance of cognitive diversity leadership and inclusion as the values of learning teams [11,12,13]. Some authors consider cognitive diversity, transformation leadership, and team work as significant aspects of innovative thinking, which is crucial in times of crises [14,15,16]. Other authors emphasize the influence of cognitive diversity in leadership on team performance [17], while emphasizing its complexity and various positive interaction. In times of crisis, the key role is the ability to make qualified decisions. The ability to make fast and competent decisions is considered the key competence by many authors [18,19]. Here we see room for exploring the link between competent crisis leadership and medical team performance, with the cognitive diversity of crisis leadership being a factor in competency that we consider important to explore, as we assume that it supports the quality of decision-making in a crisis. Based on the literature review and for the purpose of this study, we consider leadership as the competence of the leaders in a healthcare facility. The aim of this study is to examine the extent of the positive effect of leadership competence on medical teams’ performance caused by cognitive diversity of crises leadership and competent decision-making. The study is the result of research in healthcare providers, which took place in the first month of the pandemic in Slovakia (March–April 2020).

### 1.1. Crisis Leadership

The use of an appropriate leadership style is one of the most important preconditions for leaders during the acute phase. However, opinions on crisis leadership vary in the scientific literature. Haddon, Loughlin & McNally [20] have divided them into two groups. The first group clearly prefers authoritarian, slightly open and highly centralized leadership and justifies it by the need for swift action over delegation in highly critical situations [21,22]. The second group leans towards a transformational style of leadership, highlighting the need for charisma, inspirational leadership, or intellectual stimulation and vision sharing. Halverson, Murphy & Riggio [23] state that a transformational style can improve performance during a crisis by demonstrating that a transformational leader cares for the well-being of followers and provides inspiration by communicating their role in a larger mission. Charismatic leadership creates a supportive social environment during a crisis. It presupposes the application of elements of supportive, fair and personal communication [24].

James et al. [25] point to the formation of effective leadership styles under the influence of expectations that employees have from their leader. This is especially important during a crisis, as employees are key stakeholders for leaders of healthcare providers. During a crisis, employees expect the leader to successfully manage it, to easily overcome difficult obstacles. They want to be supported by him, they demand support from him, they want to feel his interest and empathy, and subsequently they want to help him through cooperation and shared leadership. Nembhard & Edmondson [26] define the term leader inclusiveness as anchoring through engagement and a sense of security in crisis situations.

Liden & Antonakis [27] a Martínez-Córcoles [28] state that each leadership must be examined in a specific context that defines the basic aspects of leadership styles. According to the authors, the most influential factor in leadership style seems to be the level of unpredictability due to the number of risk components involved in the situation and their novelty, which requires joining forces and cognitive diversity. The authors of studies focused on corporate culture found that during an internal crisis in a hierarchical organizational culture, a directive leadership style is more effective, while during an external crisis, a transformational leadership style is more advantageous [29,30]. On the other hand, clan culture or adhocracy may need a style of transformational leadership to cope with both internal and external crisis environments [29]. Nevertheless, Baran & Scott [31] point to the need to integrate directive and participatory behavior of leaders, especially in tense situations where both information and agility are required.

A literature review has identified a variety of other leadership skills that have been shown to be important during a crisis. These include, in addition to charisma, vision, communication, integrity, intelligence, authenticity, influence, emotion management, self-confidence, and participatory decision-making [32,33,34,35]. Leaders are expected to create a culture where members of the organization are motivated and rewarded for systematic thinking [32]. In a crisis context, leaders must take direct responsibility for organizing a work environment that introduces a competence-based approach to leaders in crises [32]. Martínez-Córcoles [28] and Kolbe et al. [36] report Speaking up as an important tool for leading and coordinating teams, which implies having a questioning attitude.

Pearce et al. [37] point out that notions of the appropriateness of centralized command-and-control leadership during a crisis are misleading. On the contrary, they highlight attributes such as freedom of action, trust, mutual understanding, and unity of efforts. The results will be higher morale and better resilience of employees, which they perceive as the best way to succeed in leading a crisis such as the one we are currently facing.

Based on the above literature research, we have extracted the most common items in leadership, from which we have compiled a baseline variable called crisis leadership (CL). All items that are part of CL are listed in the model in Figure 1 and Table 2.

### 1.2. Team Performance

The quality of healthcare is an essential element in achieving a high level of productivity in healthcare organizations and is defined as the extent to which the expected outcome of treatment increases in accordance with updated healthcare expertise and skills [6]. The medical institute (Institute of Medicine) has described six characteristics of high-quality healthcare performance, i.e., safe, effective, reliable, patient-centered, efficient, and equitable. 

There have been many studies published, identifying leadership style as the key element of quality healthcare service. Significant positive correlations have been noted between effective leadership style and a high level of patient satisfaction and reduction of adverse effects [4]. In addition, the literature emphasizes that effective leadership is linked to the healing process and patients‘ safety by promoting greater expertise in nursing by increasing staff stability and reducing turnover [5].

According to Sfantou et al. [2] effective leadership has an indirect impact on reducing mortality by inspiring, retaining, and supporting experienced staff. From the above, it is clear that effective leadership affects the overall outcome, i.e., the quality of healthcare by strengthening the performance of teams. Since in the initial stages of the pandemic, in which this research study was carried out, it was not yet possible to quantify the effect of leadership on performance through the final outputs of healthcare providers, thus we focused on the effect of leadership on quality of performance at the level of medical teams. 

In this study, we lean on studies that examine the impact of healthcare leadership on medical teams and these prove that openness of communication and care have a positive effect on employees’ feeling of safety and security, as well as their satisfaction [38]. In times of crises, it is important to eliminate the stress, rebuild the trust and focus on employees ‘satisfaction; and positively incentivizing their performance towards contemporary goals of healthcare facility. Based on the literature review, we assume that the quality of crises leadership (CL) is positively associated with performance of medical teams (MTP) in times of crises. 

**Hypothesis** **1 (H1).***CL is positively associated with MTP*.

### 1.3. Cognitive Diversity

Cognitive diversity has been defined as a difference in knowledge and perspective, based on professional diversity [39]. The tool for measuring cognitive diversity was introduced by Van der Vegt & Janssen [40] and this was subsequently used in several empirical studies, e.g., [16]. It explains how the members of the group differ in their way of thinking, knowledge and abilities, how differently they see the world, and their belief in what is right.

Cognitive diversity is directly related to the role or job position, especially in knowledge tasks or decision-making, and is a natural feature of every team [41]. According to van der Vegt, Bunderson and Oosterhof [42], there are two types of task-related cognitive diversity, namely expertise diversity and expertness diversity.

Cognitive diversity can be a competitive advantage for an organization due to the stimulation of consideration of non-obvious choices in task groups [43] and can improve the quality of decisions [13]. According to Horwitz [44], it is the diversity of perspectives that contributes to the success of the team as a whole. Further research has also confirmed that cognitive diversity should be considered to be a key factor influencing group decision-making processes that influence the way business resources are allocated and the definition of key competencies [39,45,46]. In addition, recent studies on innovation management have shown a positive effect of cognitive diversity in the context of a unit’s ability to create new innovative solutions [12]. Extensive research on cognitive diversity suggests that it can increase creativity, especially when transformational leadership and team perspective-taking are high [14,15,16]. In line with the decision-making perspective, Pieterse, van Knippenberg & van Ginkel [45] emphasize the importance of cognitive diversity in the context of increasing supplementary information in uncertain times.

However, it is also important to point out the risks of this phenomenon [46], as diversity can be the cause of a higher level of disagreement and a source of conflict in teams [47,48]. Lantz and Brav [49] state that cognitive diversity may not always generate positive effects, as perceived diversity leads to the creation of prejudices between and within teams [50]. It can also slow down decision-making due to difficulties in reaching consensus and reducing an organization’s ability to respond to changes in the environment [51].

Based on the above, we assume that cognitively diverse CL will make better decisions even in the phase of acute crisis (CDM—Crisis decision making). 

**Hypothesis** **2 (H2).***The relationship between CL and MTP is positively mediated by CDM*.

### 1.4. Decision-Making during a Crisis

An essential part of CL is the ability to make quick and informed decisions. Boin et al. [18] argue that it is in the acute phase of a crisis that decision-making and opinion forming are one of the first important responses of leadership. Helsloot and Groenendaal [19] consider the ability to make quick and informed decisions as one of the most important skills at a time when a crisis has unexpectedly erupted. Decision-making during a crisis is a much more demanding process than in non-crisis times. Due to turbulence, leaders have less time to make decisions, less information available, and increased decision-making workload [52]. The scientific literature focuses on highlighting the need for prompt and accurate decision-making. The accuracy and reliability of the results of the decision-making process are also important. Decision makers must be able to balance uncertain and incomplete scientific knowledge, ethics, and political and social realities in their decision-making. Bakonyi [52] deals with the issue of centralization and decentralization of decision-making and, based on the study, argues that the crisis increases the likelihood of centralization in decision-making. Aghion and Bloom [53] state that decentralization of decision-making increases the factor of productivity and efficiency of decisions even during a crisis. The authors argue that there is a correlation between qualified leaders and decentralization, but the direction of causality is not clear. Pearce et al. [37] also talk about the need for rapid decision-making linked to the transfer of responsibility and to a level where it can be best used. An important prerequisite for making the right decision is the ability to think critically, to perceive information in context and to be knowledgeable about the problem [54]. During a crisis, it is necessary for leaders to be able to analyze various options, it is not appropriate to implement the first option. Speed is important, but it does not mean haste or some form of inactivity [55].

Decision-making during a crisis is associated with a high degree of uncertainty. According to Hirsh, Mara and Peterson [56], uncertainty is a critical adaptation issue for any organism. In the case of the Covid-19 pandemic, the virus is largely unknown. Its spread is not linear, there is very little control over this new type of virus, many factors are even completely out of human control and thus, it increases uncertainty. All these factors significantly influence the behavior and decisions of leaders. Threat, uncertainty, and anxiety influence leaders, but also people in general, to make short-sighted decisions [55]. Therefore, Hirsch, Mar & Peterson [56] state that the adoption of clear goals and structures helps to bring certainty to great uncertainty.

Knowledge stated above is very important in times of current crises, when the major perspective needs to be human health protection, and meanwhile it is important to consider many other aspects such as economic, social, etc. The leadership of health-service providers needs to be careful, ready for the worst scenarios, and in the meanwhile ready with alternatives, which will be active in case the worst scenarios will not be needed. In this context, the ability of crises leadership to learn from current situations, be flexible in reactions, seems to be crucial for effective leaders [57]. 

We will examine the quality of decision-making by its positive effect on the performance of medical teams (MTP). Based on these theoretical literature reviews, we have extracted the most emphasized components of decision-making in crises (CDM) and cognitive diversity (CDL), which we have determined as mediation variables, i.e., crises leadership decision-making and cognitive diversity of crises leadership. All the items, which are components of both mediation variables are listed in Figure 1, Table 2. Based on these theoretical assumptions, we assume that CL will lead to the performance of medical teams (MTP) in the acute crisis phase through the cognitive diversity of crisis leadership (CDL) and the quality of leader decision-making (CDM).

**Hypothesis** **3 (H3).**
*The relationship between CL and MTP is positively mediated by CDM.*


**Hypothesis** **4 (H4).**
*The relationship between CL and MTP is sequentially and positively mediated by CDL and CDM.*


Figure 1 shows the model used to test the relationships between CL, CDL, CDM, and MTP. The model considers the mediating role of CDL a CDM in the relationship between CL and MTP.

## 2. Materials and Methods

### 2.1. Sampling

The questionnaires were sent electronically to team leaders of medical teams (head physicians, chief nurses) in healthcare facilities in Slovakia. The research included institutions from the whole territory of the Slovak Republic. They were intentionally not distributed directly to board of directors who are involved in leading the crises of healthcare facilities to avoid responses distorted by their subjective viewpoints, which are often perceived differently by their direct subordinates. To objectively assess the leadership competencies of leaders during an acute crisis, data were collected during the first month after the crisis (April 2020), whereas Covid-19 was first confirmed in Slovakia on 6 March 2020. It was during this period that leaders had to deal with many unexpected and unknown problems and to address a variety of issues ranging from health issues of the population and employees, the functioning of healthcare facilities, to the provision of staff, their safety and quality working conditions under pandemic conditions. The convenience sampling method was used to create the research sample. We have approached healthcare facilities in every region. There are hospitals in each district in Slovakia. The sample was designed to include respondents from all regions, size categories, and types. 62 tertiary healthcare providers were approached for cooperation through communication with their top management. The research aims and research design were explained to the hospital directors. The research was conducted in the phase of acute crisis, when hospitals were facing pressure and uncertainty. Therefore, it was not possible to track the response rate. In healthcare providers, whose management agreed to participate in the research, questionnaires in electronic form were distributed to medical team leaders who were respondents to the research. The research sample consisted of 216 leaders (team leaders of medical teams) of both types of hospitals, public as well as private. The detailed characteristics of the research sample are described in Table 1.

The SPSS 22 software (IBM, New York, NY, USA) package was used for data analysis. The reliability of the defined sets of items for individual variables (CL, CDL, CDM, and MTP) was tested using the Cronbach coefficient. The correlation analysis was used to test the relationships between sets of items compiled to evaluate individual variables. Subsequently, a mediator model according to Baron and Kenny [58] was used, and the Sobel test was used to test the mediator effect. Finally, the regression analysis was used to verify the proposed hypotheses. The control variables were the size of the healthcare facility according to the number of employees, gender and age of the team leader, his position in the managerial hierarchy, and the length of experience in the team leader position. ANOVA was used to analyze multiple dependence. We worked at a significance level of 5%.

### 2.2. Variable Measures

A mediator model was used to test the relationships between CL, CDL, CDM, and MTP which considers the mediating role of CDL and CDM in the relationship between CL and MTP. Through mediation, we can examine the causal relationships between variables and involve other variables in the underlying relationship to better examine the relationships and processes that take place between the identified variables.

CL is an independent, explanatory variable. This variable was operationalized as a score obtained by the leaders of a healthcare facility based on a rating of 10 items, using 5-point Likert-type scales (1 = strongly disagree and 5 = strongly agree). After reliability analysis, the Cronbach’s a of CLwas 0.973 (10 items).

The second variable, understood consequently, is the dependent team performance variable (MTP). Based on the study of Kasha et al. [59], the items identifying the performance of the team depend on the environment and the situation in which the performance is measured. Creating the right conditions leads to improved teamwork and thus to the achievement of the desired goals. In the acute crisis phase, it is not possible to measure team performance through quantitative indicators, as they are not yet available. A prerequisite for the effective functioning and performance of a team during a crisis is the creation of such conditions for its activities, which lead to measurable outcomes for future process evaluations and recommendations in the subsequent phase of a crisis. Team performance variable is operationalized as the healthcare organization score given by heads of the departments in terms of job satisfaction, sense of safety, and quality and safe working conditions. We have used the Safety Attitudes Questionnaire (SAQ), which was validated by many researchers in the healthcare environment and which was developed to identify attitudes of team leaders to teamwork issues in terms of teamwork climate, job satisfaction, perceptions of team leader, safety climate, working conditions, stress recognition even in exceptional situations, such as the current pandemic [60]. For our purposes, we have selected the items from the questionnaire and listed them in Table 2. After reliability analysis, the Cronbach’s a of MTP was 0.974 (16 items).

CDCM and CDM were identified as mediator variables, representing a transition bridge between the dependent and independent variables. They are directly linked to the relationship between the two variables and affect the whole model. The independent variable is the cause of the mediator variable, which is then the cause of the dependent variable [61]. The individual mediator variables are operationalized as the healthcare organization score given by heads of the departments to selected items, which have been extracted based on the above literature research. After reliability analysis, the Cronbach’s a of the CDL was 0.892 (4 items), CDM 0.972 (8 items).

The relationship between variables can also be affected by external, so-called control variables. For control variables, we verified their influence on the course of the basic investigated or modelled relationship. The gender and age of the team leader, the position in which he works, the length of his experience and education, the size of the healthcare facility as well as the sphere of activity (private or public) were examined as control variables.

## 3. Results

The relationships between individual variables are determined by means of a correlation matrix. To construct it, we have created summary variables—leadership competencies of a team managing the crises, the result of teamwork, cognitive diversity, and decision-making of leaders as the overall average score of the relevant items. Control variables are also included in the matrix. Descriptive statistics and the correlation matrix itself are shown in Table 3.

It is clear from the correlation matrix that there are significantly positive correlations between all four variables examined, indicating the use of a mediator model.

In mediation, we referred to the main hypothesis.

**H.** *The dependence between leadership competencies during a crisis and team performance is mediated by cognitive diversity, supporting the quality of decision-making in crisis*.

Since we have worked with two mediator variables, for which we assume a mutual relationship, we have divided the model into two parts, i.e., ways. The indirect, or mediated relationship goes through one of them, including both mediator variables in series, the other is the path for the direct relationship. We have proceeded in steps (C, A, B), in which we have verified the partial hypotheses by means of a gradual calculation of regressions.
There is a relationship between team performance (Y) and leadership competencies during a crisis (X).There is a relationship between mediator variables (M1, M2) and leadership competencies during a crisis (X).There is a relationship between team performance (Y) and mediator variables (M1, M2), in which X does not participate.
while
C represents the total effect,the mediated (indirect) effect of X on Y by M1 and M2 is expressed in the form A1 * B1 + A2 * B2 + A1 * B2 * D21, where the term D21 is the path from M1 to M2,the difference C’ = C − A1 * B1 + A2 * B2 + A1 * B2 * D21 is the pure (direct) effect of X on Y without the participation of M.

The hypothesis applies when the indirect effect is significant.

That is, if A1 * B1 + A2 * B2 + A1 * B2 * D21 = C − C’ is significant (using the Sobel test). When modelling the overall effect, we took into account the control variables age, gender, job position of the team leader, length of experience, education, and size of the healthcare facility. ANOVA was used to analyze multiple dependence. We worked at a significance level of 5%. The results obtained are shown in Table 4.

The decomposition of the variance for the overall dependence in the initial model has shown that the control variables are significant for the position of the team leader and the size of the healthcare facility, and as they influence the course of the tested relationships, the mediator effect will be treated.

The results in Table 5 clearly show that the overall indirect effect is significant in the positive direction. Since the direct effect of C is also significant and the dependence is positive, we can refer to incomplete serial mediation. More than half (52%) of the overall effect of the impact of leadership competencies during a crisis on team efficiency is mediated by mediators, i.e., the quality of decision-making, which is influenced by the cognitive diversity of leadership in crisis.

When interpreting all the obtained results, we have proceeded by the following steps (A, B, C):We have found that the relationships expressed by steps A and B are significant, so there are relationships of cognitive diversity of leadership in crisis (M1) and decision-making (M2) and leadership competencies (X) and at the same time there is a relationship between team efficiency (Y) and both mediator variables (M1, M2), in which X does not participate. As a result of the significance of these relations, a precondition for the existence of mediation arises.The product of parameters A1 * B1 + A2 * B2 + A1 * B2 * D21, where member D21 is the path from M1 to M2 is significant, so the indirect effect of leadership competencies (X) on team efficiency (Y) through decision-making, supported by cognitive diversity of leadership in crisis has been confirmed. The main hypothesis has thus been supported.Both indirect and direct effects are significant. In percentage terms, we can see that about 48% of the total effect is due to the direct effect and about 52% to the indirect effect. As the indirect effect does not reach more than 80% of the total effect, it is a partial mediation.

## 4. Discussion

The hypothesis of dependence between leadership competencies and team performance, which is mediated by the quality of decision-making, supported by the cognitive diversity, has been confirmed by research. However, partial mediation has been identified, where only part of the effect is mediated by mediator variables. The remaining but smaller part is transmitted directly. This is an important finding, namely that the team’s performance in the acute crisis phase is influenced by leadership competencies and its positive effect can be further enhanced through qualified decision-making, using a variety of knowledge. Our findings are consistent with many studies and findings presented in the scientific literature and complement the theory of CL with other contexts.

The research results published in this study are linked to current knowledge about the impact of leadership characteristics on the implementation of strategic and operational decisions, performance, and engagement of members of medical teams. They support the findings of several other research studies on leadership in healthcare providers, which demonstrate the positive impact of the quality of leadership on team performance, the positive impact of management education on medical teams [62], and the positive impact of leadership style. Studies have shown that the formation and implementation of decisions made by a team of leaders is related to their experience and professional education [63], the nature of the prevailing organizational culture [64], and leadership competence [62]. The content of leadership competence, which is positively associated with performance, refers to the knowledge of the organization of the healthcare facility, identification with goals and knowledge of current and expected results [65]. The values that come to the fore during the pandemic are humanity, trust, health, transparency, and activity [66].

The period of the crisis generally renders key characteristics more visible, highlights important connections and makes room for eliminating redundancy. The results of the research have shown that the speed of decision-making of leaders is highly valued, with the current caution and readiness for various, even the worst-case scenarios. The ability to analyze various alternatives and consider the ethical and moral implications of decisions is a major strength of decision-making in healthcare providers in the acute crisis phase. At the same time, employees appreciated that leaders were leading by example, which significantly supported their sense of security. In terms of the cognitive diversity of the team managing the crisis, the diversity of knowledge and skills that made informed decisions was most widely applied and exploited. On the other hand, value diversity was lower, which contributed to the value consistency of decisions, which is extremely important in serious situations. At the time of the acute phase of the current crisis, the employees of healthcare providers felt less support from the leaders, they were to a greater extent a source of support for each other. Nevertheless, according to their own statements, they were engaged, they cared about the successful handling of the crisis situation, they felt proud of their work and they were able to learn a lot from this experience.

Based on the results of the research, we can further state that it is important for the performance of medical teams that the leadership is an example of expected behavior during a crisis. Such behavior significantly promotes a sense of security and consistency in the decision-making of leaders during a crisis. It also promotes engagement, cohesion, mutual support, and, ultimately, a sense of pride in one’s work, even during a crisis. Superior CL enables medical teams to learn and enrich themselves with crisis experiences.

This research has implications for relevant change in legislation. In times of crisis, health-service leaders must act in line with their organizations’ strategies, while creating and promoting a culture of trust and teamwork among employees that helps organizations manage the crisis successfully. At the same time, however, employees in these positions often do not have sufficient leadership skills, most of which are acquired through experience. Investing in the development of leadership skills of medical team leaders should become an integral part of the healthcare facility strategy. Development of leadership has become not just improving the leadership skills of the individual, but is also an essential part of the development of the organization as a whole. Progressive health systems, which invest in the development of leadership skills of the entire management, will have a more significant return on investment in terms of organizational performance. It is, therefore, necessary to promote this idea creating a legislative framework for the support of education of medical team leaders in the field of leadership, diversity management, and organizational culture. Current practice, which in the conditions of the Slovak Republic allows the gaining of leadership skills during practice only, does not appear to be optimal from the point of view of the study results. On the contrary, it would be appropriate to legislatively support the access of health-service leaders in management-related education and to remove all barriers in the development of leadership skills.

## 5. Conclusions and Research Limitation

The success of healthcare providers in the form of medical and financial efficiency is the result of successfully leading the cooperation of teams of highly specialized professionals with full responsibility for human life. The crisis that has entered health services as a pandemic is first a crisis of health and safety, and secondly a crisis of efficiency and effectiveness. However, the performance of healthcare providers during a crisis is a critical determinant of success in controlling the pandemic with an impact on society as a whole. Thus, it is important to examine the elements of quality leadership in healthcare providers and their impact on the performance of medical teams. The subject of research in this study was the influence of cognitive diversity and the quality of decision-making of leadership in crises on the performance of medical teams in the acute crisis phase. The practical implications of the research are as follows: (1) Leadership during a crisis supports the performance of medical teams if a crisis-leadership team is shaped with respect to the diversity of knowledge and skills. (2) Decision-making processes using a diversity of knowledge can help to make quick decisions with targeted responsibility. (3) CL, influenced by the cognitive diversity, is a source of trust, satisfaction, and engagement in medical teams. (4) Value consistency of leadership in crisis is a source of certainty and transparency of decisions of leaders.

The results of the research in this study contain limitations that need to be stated for the purpose of interpreting the findings. Above all, it is research on a limited sample (216) of healthcare providers, all of which are in Slovakia. From a regional point of view, the results are relevant, and a sample would be needed for generalization. The research was conducted on a sample of healthcare team leaders, chief physicians, and chief nurses. Different work arrangements in healthcare providers can cause bias in defining the impact of leadership on performance.

Organizations rarely allocate adequate resources to prepare for leading in crisis, as it is very difficult to predict a crisis. Much of the knowledge in this area is at the level of theoretical or summarized conclusions and recommendations from previous crises. There are contingency plans in place, but often only formally, which many times make the shocking situation even more difficult because of their bureaucratic background. Lockwood [67] has shown why leaders and organizations fail in this regard, referring to too much reliance on weak, untested plans that do not effectively protect organizations in a real crisis, ignorance or failure to intercept warning signals in a timely manner, mitigating the situation, and rejecting an impending threat to the organization.

A retrospective look allows us to capture lessons learned. It is possible to learn a lot from the crisis, to identify limiting factors and to prepare for their elimination in the future. However, there is the phenomenon of recency bias, which is caused by individuals and teams remembering recent events and accentuating more important but earlier lessons [68]. The author notes that lessons learned are a modest value, hence, we are looking for leveraging lessons learned. A crisis can be perceived not only as a state of emergency, but also as an opportunity. It has repeatedly manifested itself in the history of humankind as a source of opportunities, intensive development of medical knowledge and technological boom. Leadership in healthcare providers during a crisis must focus on balancing safety and flexibility in decision-making, since safety is a source of satisfaction, and rapid adaptation leads to performance.

## Figures and Tables

**Figure 1 healthcare-09-00313-f001:**
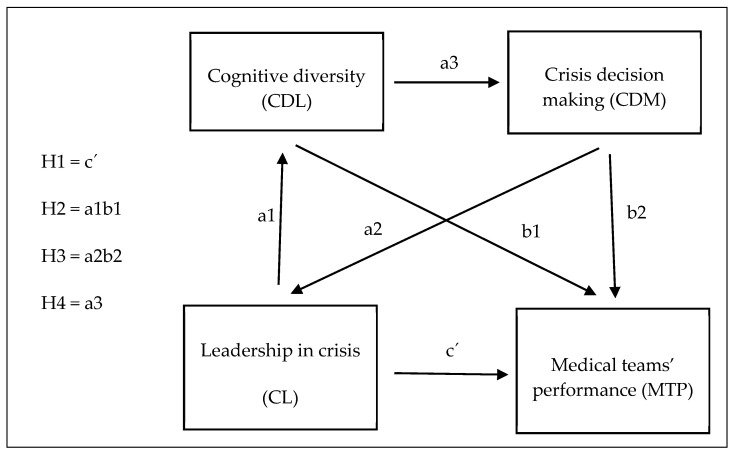
The mediation model and the 4 tested hypotheses. Source: own processing.

**Table 1 healthcare-09-00313-t001:** Structure of the examined sample of healthcare facilities. Source: authors’ results.

Variable	Category	Frequency	Percentage	Variable	Category	Frequency	Percentage
Number of employees	up to 10	10	4.6	Ownership	private	92	42.6
	11–50	24	11.1		public	124	57.4
	51 to 250	86	39.8		Total	216	100
	over 250	69	44.4				
	Total	216	100				
Position	Informed employee	54	25	Years of experience as team leader	Less than 1 year	2	0.9
	low mgmt.	46	21.3		Up to 5 years	10	4.6
	middle mgmt.	92	42.6		6 to 10 years	48	22.2
	top mgmt.	24	11.1		over 10 years	156	72.2
	Total	216	100		Total	216	100
gender	male	90	41.7	Age of leaders	up to 30 years		
	female	126	58.3		from 30 to 50 years		
	total	216	100		over 50 years		
					Total		
Education	secondary	10	4.6				
	specialized managerial	32	14.8				
	university 1st degree	6	2.8				
	university 2nd degree	124	57.4				
	university 3rd degree	44	20.4				
	Total	216	100				

Source: own processing

**Table 2 healthcare-09-00313-t002:** Items used to measure variables, source: own processing.

Crisis Leadership 1 = Strongly Disagree and 5 = Strongly Agree	Team Efficiency 1 = Strongly Disagree and 5 = Strongly Agree
Crisis leadership sets an example to its employees. Crisis leadership shows confidence in its employees even if they encounter failure. Crisis leadership provides the necessary support to employees. Crisis leadership empowers employees, provides them with room for decision-making and action if they have the necessary skills. Crisis leadership expresses support for other entities (e.g., communities, self-government, etc.). Crisis leadership puts the welfare of teams above its own interests. Crisis leadership considers the moral and ethical implications of its own decisions. Crisis leadership is optimistic about the future. Crisis leadership critically reassesses its own expectations in relation to their suitability and accuracy. Crisis leadership helps others develop their strengths.	Our facility is a good place to work even during a crisis. I am proud of the work of our facility in order to handle a crisis situation. Work in our facility is part of a large family even under crisis conditions. The morale of our (team) facility is high during the crisis. I do my job with enthusiasm even during a crisis. Currently, my work gives me enough autonomy. In the current situation, I receive useful feedback from the team leader. All the necessary information for diagnosis and therapeutic decisions is currently available to me. The working environment of our facility is safe during a crisis. Working conditions during a crisis are satisfactory for our facility. Team leaders deal constructively with the problems of their subordinates during a crisis. Team members help and support each other when working in the current situation. Team members are willing to work harder during a crisis. I care about how successfully our facility handles a crisis situation. Despite the crisis situation, my work is a source of energy for me. I learn a lot in my work even in this crisis period.
**Cognitive diversity** **1 = strongly disagree and 5 = strongly agree**	
I assume that individual people who are part of leadership in crises are different from each other in the way of thinking, their knowledge and skills, the way they see the world, their beliefs about what is right or what is wrong.	
**Decision-making during a crisis 1 = strongly disagree and 5 = strongly agree**	
Leaders are knowledgeable about the problem during a crisis. Leaders’ decisions are in line with the strategy or vision or values even during a crisis. Leaders’ decisions are quick during a crisis and leaders take responsibility for them. Leaders are able to critically evaluate information during a crisis. Leaders are able to perceive information in context during a crisis. Leaders are able to analyze various possible solutions during a crisis. Leaders are able to learn on the go from situations during a crisis. Leaders are careful during a crisis and is ready for the worst-case scenarios.	

Source: own processing

**Table 3 healthcare-09-00313-t003:** Correlation matrix. Source: authors’ results.

Variable	Mean	SD	CL	CDM	CDL	MTP	Age	Gender	Position	Experience	Size	Education
CL	3.39	1.14										
CDM	3.64	1.12	0.940 **									
CDL	3.73	0.97	0.395 **	0.355 **								
MTP	3.66	1.06	0.891 **	0.888 **	0.501 **							
Age	2.22	0.53	0.250 **	0.228 **	0.079	0.243 **						
Gender	1.58	0.49	0.351 **	0.320 **	0.005	0.316 **	0.035					
Position	2.40	0.98	0.314 **	0.329 **	−0.018	0.186 **	0.203 **	−0.040				
Experience	3.66	0.61	0.264 **	0.245 **	0.271 **	0.263 **	0.518 **	0.049	0.212 **			
Facility Size	2.24	0.83	−0.044	−0.047	−0.047	0.029	−0.058	0.163 **	−0.038	0.163 **		
Education	4.56	1.44	−0.180 **	−0.142 **	−0.038	−0.204 **	−0.052	−0.092	0.145 **	−0.079	−0.144 **	
Sphere	1.57	0.494	−0.257 **	−0.253 **	−0.352 **	−0.268 **	−0.238 **	0.032	0.140 **	−0.115 **	0.410 **	0.046

Note. LC = leadership competencies of leaders; CDM = decision-making in crisis; CDL = cognitive diversity of team managing the crises; MTP = medical team performance; ** *p* > 0.05. Age: 1—up to 30 years, 2—30–50 years, 3—over 50 years; gender: 1—female, 2—male; positions: 1—informed employee (employee with responsibility to lead medical team, without formal management position in organizational hierarchy), 2—low management, 3—middle management, 4—top management; experience: 1—less than a year, 2—up to 5 years, 3—up to 10 years, 4—more than 10 years; facility size: 1—up to 10 employees, 2—11 to 50, 3—51‒250, 4—over 250; education: 1—secondary, 2—specialized managerial, 3—specialized HE 4—university 1st degree, 5—university 2nd degree, 6—university 3rd degree; sphere: 1—private, 2—state. Source: own processing.

**Table 4 healthcare-09-00313-t004:** Tests of Between-Subjects Effects. Source: authors’ results.

Source	Type III Sum of Squares	df	Mean Square	F	Sig.
Corrected Model	199,192	16	12,449	57,616	0.000
Intercept	1096	1	1096	5071	0.025
Sphere	0.281	1	0.281	1300	0.256
Age	0.054	1	0.054	0.249	0.618
Gender	0.052	1	0.052	0.243	0.623
Position	2111	1	2111	9768	0.002
Experience	0.009	1	0.009	0.040	0.842
Size	1484	1	1484	6868	0.009
Education	0.003	1	0.003	0.012	0.913
Leadership	112,583	1	112,583	521,036	0.000
Error	42,999	199	0.216		
Total	3,138,350	216			
Corrected Total	242,191	215			

Source: own processing.

**Table 5 healthcare-09-00313-t005:** Parameter Estimates. Source: authors’ results.

**Model 1/Step C**						
**Dependent Variable: Result**						
**Parameter**	**B**	**Std. Error**	**t**	**Sig.**	**95% Confidence Interval**	
					**Lower Bound**	**Upper Bound**
Intercept	0.820	0.145	5.651	0.000	0.534	1.106
Leadership	0.861	0.029	29.202	0.000	0.803	0.919
**Model 2/Step A**						
**Dependent Variable: Cognitive Diversity of Leadership in Crisis**						
**Parameter**	**B**	**Std. Error**	**t**	**Sig.**	**95% Confidence Interval**	
					**Lower Bound**	**Upper Bound**
Intercept	3.005	0.280	10.725	0.000	2.453	3.557
Leadership	0.344	0.057	6.042	0.000	0.232	0.456
**Model 3/Step A**						
**Dependent Variable: Decision-making**						
**Parameter**	**B**	**Std. Error**	**t**	**Sig.**	**95% Confidence Interval**	
					**Lower Bound**	**Upper Bound**
Intercept	0.365	0.147	2.479	0.014	0.075	0.656
Leadership	0.904	0.026	34.619	0.000	0.853	0.955
**Model 4/Step B**						
**Dependent Variable: Result**						
**Parameter**	**B**	**Std. Error**	**t**	**Sig.**	**95% Confidence Interval**	
					**Lower Bound**	**Upper Bound**
Intercept	0.017	0.155	0.107	0.915	−0.289	0.322
Leadership	0.414	0.070	5.922	0.000	0.276	0.552
Cognitive Diversity	0.205	0.030	6.791	0.000	0.146	0.265
Decision-making	0.412	0.071	5.776	0.000	0.271	0.552
	Indirect effect					
A1 * B1	0.071					
A2 * B2	0.372					
A1 * B2 * D21	0.004					
Indirect	0.447					
z	6.652					
Sig.	0.000					
Effect		Coefficient			%	
Total		0.861			100	
Direct		0.414			48	
Indirect		0.447			52	

Source: own processing.

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
