# Peer review of "Cognitive Diversity as the Quality of Leadership in Crisis: Team Performance in Health Service during the COVID-19 Pandemic"

_healthcare, 2021, doi:10.3390/healthcare9030313_

Round 1

Reviewer 1 Report

First paragraph extremely long and could flow better. Maybe start with "Published...(line 42) and combine and condense the material in the three current first sentences. Too many concepts with inadequate connections between them Either unpack more and sequence so the relationships aare clear and/or condense/combine/eliminate some. Suggest a break at line 62.

"Managerial competence depends on the mana-50 gerial skills of managers, which are often underestimated and are the source of many con-51 troversial discussions about their inadequacy [6–9]." --really? Does it not depend on many things? If o it seems the problem would be their being OVERestimated. Clarify.

Line 166 sounds like authors are claiming to have coined/invented crisis leadership, but it exists in literature.

Similarly first paragraph of 1.2 is long.

Line 211--what is the "this"--cognitive diversity? Restate for clarity.

Several instances of subject-verb non-agreement and similar grammar issues

Author Response

Thank you for your valuable comments, we have adjusted the text taking all of them into consideration.

  1. First paragraph extremely long and could flow better. Maybe start with "Published...(line 42) and combine and condense the material in the three current first sentences. Too many concepts with inadequate connections between them Either unpack more and sequence so the relationships aare clear and/or condense/combine/eliminate some. Suggest a break at line 62.

We have done major revisions in the theoretical background based on the reviewer’s recommendations. We have reference has been compressed to avoid repetition in case of similar results of various studies.  

  1. "Managerial competence depends on the mana-50 gerial skills of managers, which are often underestimated and are the source of many con-51 troversial discussions about their inadequacy [6–9]." --really? Does it not depend on many things? If o it seems the problem would be their being OVERestimated. Clarify.

We have more focused on being specific on the perception of management and leadership used in the study. We have explained the issue of discrepancy between the attention drown to leadership in healthcare service in the research and in practical application (line 49-56).

  1. Line 166 sounds like authors are claiming to have coined/invented crisis leadership, but it exists in literature.

We have lean on many studies clarifying the content of crisis leadership and competence involved. We consider crisis leadership under conditions of the pandemic valuable subject for the research. Our aim was to explore, if and how can the quality crisis management, supporting performance in pandemic, be supported by leadership competence. We find the research valuable, because it was done right at the beginning of the pandemic, where there was not much experience in health service, how to lead medical teams through pandemic crisis.

  1. Similarly first paragraph of 1.2 is long.

We have structured the text to support logic in choice of the mediation model and its variables.

  1. Line 211--what is the "this"--cognitive diversity? Restate for clarity.

We have reformulated the text, so there is more clearing meaning. (line 249-255)

  1. Several instances of subject-verb non-agreement and similar grammar issues

We have had the grammar revised with experienced reviewer.

Reviewer 2 Report

Two things bother me about this article. The first is the apparent confusion in terminology between leadership and management in the introduction, literature review and research design ... these are not interchangeable terms but are commonly confused, which should not happen in a peer-reviewed journal. The second is the lack of an introductory definition of "cognitive diversity", based on literature. There is no argument that - if these were remedied - the article would improve in value. But I worry that the confusion in terminology has also confused the research design and even the questions. I recommend that the authors be requested to revise the first half of the article to ensure and demonstrate that the research design captures leadership behaviour, not management behaviour. Also, crisis and disaster studies are well enough advanced to provide a definition, even though the authors suggest not. Persevere with this project and it will be well received by our audience.

Author Response

Thank you for your valuable comments, we have adjusted the text taking all of them into consideration.

  1. Two things bother me about this article. The first is the apparent confusion in terminology between leadership and management in the introduction, literature review and research design ... these are not interchangeable terms but are commonly confused, which should not happen in a peer-reviewed journal. I recommend that the authors be requested to revise the first half of the article to ensure and demonstrate that the research design captures leadership behaviour, not management behaviour.

Thank you for pointing out that in formulating the text, we have not been able to make it clear that our intention was to focus on crisis leadership and the content of leadership competence in times of crisis. We consider crucial to explore the conent of leadership supporting performance in pandemic

We have done major revisions in the literature review, we have structured the text and also supported the study with more sources referring to leadership competence and management content in crises. To avoid unclear terms, we have made changes in labelling the variables in mediation model as well as the title of the paper. The leadership in crises may be realized in several ways. As the tested items in the questionnaire show, we have been examining the leadership from the perception of trust and autonomy support towards employees.

  1. The second is the lack of an introductory definition of "cognitive diversity", based on literature.

We have added chapter with the literature review of cognitive diversity, where we have explained its content and benefits for organizations. We have found this recommendation relevant and helpful to the logic of the text, thus we have decided to support all the variables in the mediation model with separate chapter in the literature review.

Round 2

Reviewer 2 Report

I have read the entire manuscript again and there is much improvement, especially in the introduction and literature review. However, there is still much confusion about the terms leadership and management, on lines 38, 57, 65-66, 81, 82, 114, 137, 165, 194, 219, 223, 233, 237, 245, 246, 255, 264, 287, 290, 293, 295 and 299, as well as in the discussion and conclusions. I gave up numbering them after Line 299. I have highlighted these in the attachment. Fig 1 is clear but Table 1 confuses the terms; Table 2 is clear at the top but confuses the terms at the bottom. The sentence from Line 203-206 is repeated on line 208-212. The confusion between the terms leadership and management undermines the whole study. Sometimes you use the word "coping with" (line 144) or "coordinating" and these work well. You have introduced the term "performance" and this works well. But scholars of leadership will worry that you don't have a clear understanding of the field ... when you probably do but the English has spoiled the work. The introduction of the definition and discussion of cognitive diversity improves the work greatly. I agree that sometimes the concept of management should be included but not in ways that clearly confuse this work of coordinating, with the work of leadership and decision making.

Author Response

Dear madam/sir,

thank you so much for the very careful review of our paper. We truly apologise for the confusion caused by the terms leadership and management. It was result of errors in professional translation. We have involved experienced linguists to resolve any linguistic confusion.  

 We are now using the term leadership in crises for people who are in charge of leading medical teams in times of crises and leadership as competence. Our research was conducted in the acute phase of the pandemic, when there wasn’t knowledge about what it takes to lead medical teams in the crisis under pandemic conditions. Therefore, the intention of the research was focused on the  leadership in crises and understanding the impact on performance of medical teams. It is very important to us to avoid any confusion and unclear terminology in our research. We truly believe that the study should bring value to current knowledge in leadership in healthcare services. 

Round 3

Reviewer 2 Report

You've done it, my hearty congratulations. I support publication and look forward to seeing this article circulated widely.

Author Response

Dear reviewer, thank you for your patience and precision. Your comments and remarks has helped us to improve the paper and meet the journal criteria.